# Bibliometric Analysis of Hydrocarbon Bioremediation in Cold Regions and a Review on Enhanced Soil Bioremediation

**DOI:** 10.3390/biology10050354

**Published:** 2021-04-22

**Authors:** How Swen Yap, Nur Nadhirah Zakaria, Azham Zulkharnain, Suriana Sabri, Claudio Gomez-Fuentes, Siti Aqlima Ahmad

**Affiliations:** 1Department of Biochemistry, Faculty of Biotechnology and Biomolecular Sciences, Universiti Putra Malaysia, Serdang 43400, Malaysia; howswen96@gmail.com (H.S.Y.); nadhirahairakaz@gmail.com (N.N.Z.); 2Department of Bioscience and Engineering, College of Systems Engineering and Science, Shibaura Institute of Technology, 307 Fukasaku, Minuma-ku, Saitama 337-8570, Japan; azham@shibaura-it.ac.jp; 3Department of Microbiology, Faculty of Biotechnology and Biomolecular Sciences, Universiti Putra Malaysia, Serdang 43400, Malaysia; suriana@upm.edu.my; 4Department of Chemical Engineering, Universidad de Magallanes, Avda, Bulnes, Punta Arenas 01855, Chile; claudio.gomez@umag.cl; 5Center for Research and Antarctic Environmental Monitoring (CIMAA), Universidad de Magallanes, Avda, Bulnes, Punta Arenas 01855, Chile; 6National Antarctic Research Centre, B303 Level 3, Block B, IPS Building, Universiti Malaya, Kuala Lumpur 50603, Malaysia

**Keywords:** bioremediation, cold region, soil, hydrocarbon, bacteria

## Abstract

**Simple Summary:**

Anthropogenic activities in cold regions require petroleum oils to support various purposes. With the increased demand of petroleum, accidental oil spills are generated during transportation or refuelling processes. Soil is one of the major victims in petroleum pollution, hence studies have been devoted to find solutions to remove these petroleum hydrocarbons. However, the remote and low-temperature conditions in cold regions hindered the implementation of physical and chemical removal treatments. On the other hand, biological treatments in general have been proposed as an innovative approach to attenuate these hydrocarbon pollutants in soils. To understand the relevancy of biological treatments for cold regions specifically, bibliometric analysis has been applied to systematically analyse studies focused on hydrocarbon removal treatment in a biological way. To expedite the understanding of this analysis, we have summarised these biological treatments and suggested other biological applications in the context of cold conditions.

**Abstract:**

The increased usage of petroleum oils in cold regions has led to widespread oil pollutants in soils. The harsh environmental conditions in cold environments allow the persistence of these oil pollutants in soils for more than 20 years, raising adverse threats to the ecosystem. Microbial bioremediation was proposed and employed as a cost-effective tool to remediate petroleum hydrocarbons present in soils without significantly posing harmful side effects. However, the conventional hydrocarbon bioremediation requires a longer time to achieve the clean-up standard due to various environmental factors in cold regions. Recent biotechnological improvements using biostimulation and/or bioaugmentation strategies are reported and implemented to enhance the hydrocarbon removal efficiency under cold conditions. Thus, this review focuses on the enhanced bioremediation for hydrocarbon-polluted soils in cold regions, highlighting in situ and ex situ approaches and few potential enhancements via the exploitation of molecular and microbial technology in response to the cold condition. The bibliometric analysis of the hydrocarbon bioremediation research in cold regions is also presented.

## 1. Introduction

In recent years, oil pollutants have significantly contaminated cold climate regions, raising problematic environmental concerns. Generally, petroleum oil has been recognised as a priority pollutant by the US Environmental Protection Agency due to its carcinogenic and mutagenic properties that have an adverse impact on the ecosystem [1]. The consumption of petroleum oil in cold regions is contributed by the elevated human activity with targeted political, social, economic and scientific interests [2]. These events have increased the probability of accidental oil leakages or spillages from the storage tanks, pipelines, industrial sites and military bases [3]. Consequently, oil pollution in cold environments (i.e., Arctic, Antarctica and other cold climate regions) can be easily found in surface soils, deep-sea waters, marine sediments and permafrost [4,5,6,7]. The harsh environments, such as the cold temperature, insufficient nutrients, low pollutant bioavailability and the tedious freeze–thaw activities, under cold regions has led to a higher vulnerability towards the petroleum pollutants compared to the tropical and temperate environments [8]. These factors also contribute to a lower rate of natural attenuation in cold regions, thus allowing the petroleum pollutants to persist in the environment for approximately more than 20 years [9]. Therefore, the requirement of clean-up strategies is essential in recent years to treat the hydrocarbon-polluted cold soils.

Physical, chemical and thermal treatments (i.e., soil excavation, permeable reactive barriers, liquid–liquid extraction, thermal desorption and electrokinetic) can be applied to eradicate the hydrocarbon-contaminated sites of different sources, including sea water, soils and sediments [10,11,12]. However, these methods usually require high maintenance costs, complex operational skills and pose harmful threats to the cold climate ecosystem [13,14]. A study by Naseri et al. [9] further clarified that these treatments were hardly applicable in the Arctic due to factors of transportation, extreme temperatures and absence of sophisticated infrastructure. On the other hand, microbial bioremediation has been proven innovative for remedying petroleum hydrocarbons from a contaminated area since it is environment-friendly, cost-effective and has an ex situ/in situ applicability [15,16]. The emerging interest of hydrocarbon bioremediation treatments requires the biotechnological improvements (bioaugmentation and/or biostimulation) to enhance their removal efficiency. These enhanced bioremediation treatments are widely applied to remedy petroleum-polluted sites in cold regions [17,18,19].

Most of the former reviews addressed the bioremediation in tropical and temperate environments, however, there are limited reviews concentrated on the bioremediation specifically in cold environments. Thus, this review updates the gap in the literature on enhanced bioremediation studies towards petroleum-polluted soils in cold regions and conceptualises their research trend by bibliometric analysis to identify the research focus. The potential applications of molecular and microbial technology to tackle various limitations are also discussed.

## 2. Petroleum Oil Bioremediation in Cold Regions

To date, cold regions including the Arctic, Antarctica, Alaska, Canada, Russia, Chile and Norway have been reported with the oil spillage events, leading to the acute contamination of their environments [20,21,22,23,24]. From the previous analysis, nearly 920 petroleum-polluted spots were found in Alaska, while the Arctic was recorded with approximately 377 sites, followed by Antarctica with roughly 200 locations [21,22,23]. Furthermore, Russia, Greenland, Iceland, Sweden, Norway and Finland, with more than 100 areas, have been reported with the existence of petroleum oil pollutants [20,23]. The general type of pollutants released from petroleum oils are summarised in Table 1. 

In June 2020, there was an accidental oil spillage (approximately 15,000 tonnes of fuel into a river and 6000 tonnes into the soil) in the Arctic Circle due to collapsed fuel containers at a power plant located in the Siberian city of Norilsk, Russia [24]. This accident was the second-largest oil spill recorded in Russia, where the crude oil drifted about 12 km from the accidental site and subsequently contaminated the Ambarnaya river with a size of 350 km^2^. The oil spill has raised a state of environmental emergency which poses a challenge to clean up under harsh conditions in the Arctic Circle. An accident similar to Russia’s, the Exxon Valdez oil spill in Prince William Sound, Alaska, was one of the historical oil spills documented in the world [33]. From the investigation, roughly 41,000 m^3^ of crude petroleum was released, contaminating the 2000 km coastline [34]. Despite the large amount of oil released, hydrocarbon bioremediation had been implemented to treat the oil spillage successfully, serving as a pioneer biological removal study in cold environments [35].

### 2.1. Hydrocarbons Bioremediation

Hydrocarbon bioremediation strategies are sustainable processes that offer low operational costs and generate little or no harmful effects to the treatment sites compared to the non-biological treatments. In general, Anjum et al. [36] stated that the high-cost efficiency (massive savings of about 65–85%) can be achieved by implementing biological treatments to remove environmental pollutants. A genuine example is the sole water cleaning of petroleum-polluted rocks from the Exxon Valdez site, which cost approximately USD 1 million per day [37]. In other words, the USD 1 million was only sufficient to treat polluted rocks with water on a daily basis, while the transportation and other physical treatment fees are not included within the 1 million US dollars. However, the implementation of bioremediation approaches required less than USD 1 million to successfully clean up a few hundred kilometres of polluted Exxon Valdez coastlines [37]. Thus, bioremediation was preferred to remove oil pollutants in the Exxon Valdez site owing to the affordable operational cost with beneficial outcomes. Nevertheless, the state-of-knowledge for the cost evaluation on bioremediation in cold regions is underdeveloped and no study has yet focused on the recent remediation cost in these regions pertaining to the huge fluctuation of the global economy in the past 10 years. Thus, more clarifications are greatly needed to estimate the genuine operational cost for the petroleum bioremediation under realistic cold environments.

Although bioremediations are potentially cost-effective and eco-friendly, such treatments in cold regions usually take more time to meet the clean-up standards owing to a few environmental restrictions, such as low nutrient bioavailability, cold temperature, low oxygen and improper soil humidity [9,38]. Hence, biotechnological improvements using biostimulation and/or bioaugmentation can be modified to enhance the TPH (Total Petroleum Hydrocarbon) remediation efficiency in response to these restrictions. Biostimulation optimises the environmental conditions needed for microbial degradation to remove pollutants. Meanwhile, bioaugmentation techniques employ inoculum of indigenous, hydrocarbon-degrading microorganisms to treat polluted soil by enhancing the biodegradation activity. Bioremediation approaches that apply biostimulation and/or bioaugmentation improvements are referred to as enhanced bioremediation.

#### 2.1.1. Biostimulation

Under cold regions, biostimulation is preferred to improve the hindered metabolic activity of autochthonous microorganisms. Autochthonous microorganisms are native microbes that are capable of using hydrocarbons as their sole energy sources for growth. Biostimulation enhances bioremediation efficiency by adding nutrients, water and aeration in polluted soils. The soil retrieved from cold environments is commonly associated with low concentrations of nitrogen (N) and phosphorus (P) that make them not suitable for the growth of microbial activity [39]. The N and P bioavailability in soil can be improved by applying natural or synthetic fertilisers whether it is an organic or inorganic compound [40]. The positive effect of nutrient additives has been highlighted in a study by Kim et al. [41], where the utilisation of humate and commercial nutrient additives (i.e., 20% N: 20% phosphoric anhydride: 20% potassium oxide) generates an efficient TPH removal despite seasonal freezing restrictions during winter. These nutrient additives promoted the growth of microorganisms by stimulating nucleic acids and amino acids production involved in cell metabolic pathways [41].

Surfactant foams containing nutrient additives and bioaugmented microbial strains that act as transferring and insulating media can be sprayed on soil surfaces in bioremediation treatment, as suggested by Jeong et al. [42]. In the study, higher microbial activity in cold conditions was observed as the foam concentrates the nutrients to support the growth of microbes and keep the soil 2 °C warmer than the control [42]. This simple foam-spraying technique can potentially reduce the exploitation of expensive material, electrical energy and manpower in bioremediation treatment under cold regions. Besides nutrients additives, easily degradable substrates such as municipal composts and organic sludges can be applied as an initial bio-stimulant in bioremediation treatments under cold climate regions [43,44]. These substrates represent a primary nutrient supply to promote the growth of autochthonous microorganisms while additive fertilisers serve as the secondary substrates for the biodegradation of hydrocarbon pollutants in contaminated soils.

The low oxygen level in contaminated soils suppresses aerobic biodegradation of hydrocarbon pollutants, giving a lower removal efficiency [45]. This limitation can be solved by supplying oxygen into the polluted soil and followed by uniform soil mixing to ensure all microbes receive the supplied oxygen. Different designed aeration structures (micro-injection point, rototilling, aggressive and periodic tilling) with an optimised airflow rate have been proposed in bioremediation treatment to clean-up hydrocarbons under cold conditions [30,45,46,47]. Soil temperature is another parameter to be assessed prior to the bioremediation implementation. Studies highlighted that the hydrocarbon bioremediation in cold regions should be implemented during the summer season, which generally has a higher temperature (10–15 °C) that stimulates the microbial activity and the thawing of frozen soils [48]. The correlation between sunlight exposure and soil temperature in cold environments has been investigated, suggesting that a longer exposure period can boost the soil temperature and enhance hydrocarbon biodegradation [49]. Thus, the hydrocarbon treatment site in cold regions is recommended to implement at locations that have great access to sunlight during summer.

#### 2.1.2. Bioaugmentation

Cold regions favour the usage of indigenous microorganisms in the removal treatment. These native microorganisms are cold-adapted, non-invasive to the treatment site and long-term staying in cold environments, which suited them to the local nutrient bioavailability and environmental conditions [50]. Laws and regulations such as the Antarctic Environmental Treaty and Arctic Environmental Protection Strategy have been proposed and enforced to conserve the ecosystem [51,52]. Under these laws, cold regions such as Norway, Sweden, Iceland and Antarctica forbid the use of foreign microorganisms to maintain their native biodiversity [51,52]. Indigenous microorganisms that colonise in soils are identified as either psychrophilic- (optimal growth below 20 °C) or psychrotolerant-typed (optimal growth of 20–30 °C) [53]. Both types are efficient hydrocarbon remediators, involving bacteria, fungi and yeast. Studies have identified bacteria as having the major remediator role, while limited cold-adapted fungi and yeasts are reported to possess petroleum biodegradation capability (Table 2). 

According to the 16S rRNA analysis, a significant microbial shift in hydrocarbon-polluted soils has been observed, involving the growth of linear- and aromatic-degrading bacteria such as the members of *β-Proteobacteria*, *γ-Proteobacteria*, *α-Proteobacteria* and *Actinobacteria* [59]. These bacteria are referred to as hydrocarbonoclastic bacteria (HCB) that can utilise hydrocarbons as their sole energy sources. Many cold-adapted HCB were identified, such as species of *Arthrobacter*, *Rhodococcus, Pseudomonas, Sphingomonas* and *Pseudoalteromonas,* that can be applied to treat polluted soils (Table 2). Studies highlighted the ability of these microorganisms to degrade petroleum-based contaminants at a lower temperature (<10 °C), suggesting their potential usage in response to the cold condition. The hydrocarbon-degrading ability of these microbes at a lower temperature are mainly attributable to the cold-tolerant enzymes and anti-freeze proteins that can catalyse essential microbial growth and degradation processes [60,61].

Besides single bacteria strain, microbial consortium can be employed to treat contaminated soils. This technique requires an integrated bacterial population that possess superior removal ability. Different bacterial species degrade different hydrocarbons effectively, such as aliphatic (linear, branched, or cyclic alkanes) or aromatic compounds (mono- or poly-aromatic hydrocarbons). For instance, Margesin et al. [25] had demonstrated the difference of removal efficiency for each strain at a low temperature. The study reported that *Rhodococcus cercidiphyllus* BZ22 degraded n-alkanes C12–C22 effectively, while other hydrocarbons were degraded significantly by *Arthrobacter sulfureus* BZ73 (phenol) and *Rhodococcus erythropolis* BZ4 (anthracene and pyrene), respectively [25]. The outcomes validate the theory of different strains having different hydrocarbon preferences. The inoculation of effective degraders into an integrated consortium potentially increases the hydrocarbon bioremediation efficiency due to the wider catabolic activity and diverse enzymatic capability. A comparative study using crude oil reported that the oil removal by *Pseudomonas* sp. BPS1-8, *Bacillus* sp. IOS1-7 and *Pseudomonas* sp. HPS2-5 were 69%, 45% and 41%, respectively [62]. However, a mixed consortium using these three strains displayed a maximum of 77% removal efficiency, which is higher than the single-strain removal study. The findings clearly demonstrated that the addition of effective hydrocarbon degraders in a mixed consortium can generate higher remediation efficiency.

Although bioaugmentation enhances the treatment efficiency on a laboratory-scale, this practice should be accessed and optimised prior to the genuine field treatment. Bioaugmented strains and consortia are influenced by biotic and abiotic factors. The limiting abiotic factors are the pollutant nature, soil temperature, pH, agitation and nutrient bioavailability, while the influencing biotic factor is the competition between each bacterial strain after mixing into a population [63]. An unfavourable environmental condition hinders the bioaugmentation efficiency. A study by Ruberto et al. [64] reported that the bioaugmentation approach had no significant effect on the TPH remediation (removal rate and concentration) in Antarctic soils, resulting in the waste of energy and resources. The study suggested that beneficial bioaugmented treatment can be achieved only with favourable environments that stimulate the growth of these microbes to degrade pollutants [64]. In other words, bioaugmentation is highly dependent on biostimulation techniques to establish an optimum environment that can stimulate microbial activity and hydrocarbon biodegradation.

### 2.2. Biodegradation Pathway and Its Metabolic Aspects

Cold-adapted bacteria can generate energy or essential metabolites by degrading petroleum hydrocarbons for biomass development. These bacteria react differently toward several types of hydrocarbons. The water-soluble alkanes with shorter length can be absorbed readily into bacteria cells for biodegradation; meanwhile, the absorption of medium- and high-molecular-weight (MLMW) hydrocarbons is facilitated by the surface-active biomolecules such as biosurfactants and bioemulsifiers [65]. These emulsifying agents decrease the surface tension of MLMW hydrocarbons by forming micelles. The micelle structure enhances their water solubility, thereby enabling the transportation of these less water-soluble hydrocarbons into the cold-adapted bacteria, passively or actively (with adenosine triphosphate (ATP)) [65]. After these substrates enter the bacterial cell, enzymatic breakdown of hydrocarbons is carried out to generate cellular metabolites for microbial growth. Interestingly, the biodegradation of petroleum hydrocarbons by cold-adapted bacteria is similar to the biodegradation in mesophilic bacteria [66]. The major difference between them is the rate of reaction, in which the former group has a slower rate compared to the latter group due to the environmental constraints [38,66]. In general, petroleum hydrocarbons can be degraded aerobically or anaerobically. However, limited data has been reported for anaerobic biodegradation compared to the aerobic biodegradation that has been highlighted for a longer period in cold conditions [38]. Thus, the general overview of aerobic biodegradation of hydrocarbons in cold environments is illustrated in Figure 1.

Aerobic biodegradation acquires oxygen as an electron acceptor that attacks hydrocarbons and initiates enzymatic breakdown [67]. Both aliphatic and aromatic hydrocarbons can be degraded aerobically with the presence of dioxygenase enzyme. For aliphatic hydrocarbons (i.e., n-alkanes), terminal or subterminal oxidation of methyl group are conducted to produce primary or secondary alcohol [65]. Primary alcohol is further dehydrogenated into carboxylic acid or fatty acid that undergoes β-oxidation to generate acetyl coenzyme A (CoA) molecules [65]. Meanwhile, the secondary alcohol generated from subterminal oxidation is converted into ketones and esters, which are broken down into primary alcohol and acetate afterward. The primary alcohol then undergoes dehydrogenation and β-oxidation to generate acetyl CoA [67]. These intermediate metabolites are essential molecules that enter the TCA (tricarboxylic acid) cycle for biomass production. In some circumstances, the monocarboxylic acid derived from primary alcohol can undergo ɷ-hydroxylation of methyl group to form dicarboxylic acid that incorporates into β-oxidation reaction. Nevertheless, the biodegradation cycle is repeated to oxidise the hydrocarbon chain until it is fully converted into carbon dioxide (CO_2_) and water (H_2_O) [65,67]. The dehydrogenation and β-oxidation reaction also produce electrons that enter the TCA cycle to stabilise cofactor molecules such as nicotinamide adenine dinucleotide, NADH.

Besides aliphatic hydrocarbons, dioxygenase oxidation of aromatic hydrocarbons (i.e., BTEX and naphthalene) generate cis-dihydrodiols that are dehydrogenated into catechol [60,67]. Catechol is the by-product of aromatic hydrocarbon biodegradation which undergoes ring fission to generate various cellular metabolites that can be incorporated into the TCA cycle [67]. Two degradation processes of catechol are reported and referred to as orthro- and meta-cleavage. In orthro-cleavage, the catechol molecules are cleaved between two hydroxyl groups with the formation of cis, cis-muconic acid [60]. On the other hand, the fission between hydroxylated and non-hydroxylated carbon in catechol is referred to as meta-cleavage that produces semialdehyde molecules [60]. These intermediate molecules are eventually converted to TCA cycle metabolites for cellular growth and maintenance.

### 2.3. Bioremediation Research Trend in Cold Regions

A bibliometric analysis was conducted to comb out the themes that propel research into the field of biotechnology and bioremediation of petroleum pollution in cold regions of the planet. Bibliometric data can suggest the developmental status and hot-topic trends [68]. Data-mining was carried out using VOS viewer (Centre for Science and Technology Studies, Leiden University, Leiden, The Netherlands) version 1.6.16 analysis software (https://www.vosviewer.com, 22 March 2021) developed by Van Eck and Waltman [69]. A thematic search was done using the search terms “biotechnology OR bioremediation AND oil pollution OR hydrocarbon AND cold OR low temperature OR permafrost OR polar OR Antarctic OR Antarctica OR Artic” in the Scopus database (https://www.scopus.com/, 22 March 2021). The search generated 32 documents consisting of review articles, research articles, editorials, mini reviews, short communications, data articles, book chapters and conference abstracts.

An analysis of co-occurrence of keywords is illustrated in Figure 2 to map the strength of association between keywords in contextual data [70]. This mapping method can estimate the similarity according to strength of association or co-occurrence. A higher co-occurrence generally resorts to measuring similarity between keywords. Larger number of publications where two keywords co-occur are indicative of the similarity of the two keywords [71]. Keyword occurrence can also be analysed via cluster mapping. It can act as an auxiliary support for scientific research and effectively deduce the hotbed of research topics in the discipline field of the research [72]. Keywords can be analysed to identify popular topics in research of microbial bioremediation of petroleum hydrocarbons in cold regions. A full count was done on the keywords which produced 4283 keywords. The minimum number of a keyword occurrence was met with a threshold of 20 which was selected to shortlist the keywords to only 81. Keyword here is defined as words used more than 20 times in titles and abstracts across all 32 documents.

Network visualisation provides an overview of the analysis of keyword co-occurrence. Three parameters were available for the analysis: link (L), total link strength (TLS) and occurrence (O). A link (L) strength between keywords is the frequency of co-occurrence. Each L is represented by a positive numeral value called a strength [73], whereby high values corelate with higher strength. TLS indicates the number of publications that occur between any two keywords [69]. Figure 2 was generated using VOS viewer software, measured in terms of TLS. Each keyword is represented by a circle. The size of the circle represents the weight, in this case measured by TLS. The bigger the circle, and word, the higher the TLS. Each circle is joined together by a line, where the thicker the line, the more co-occurrence they have [74]. The location of any given circle in the cluster map is determined based on the specific point in time of source publication and its relationship to other circles [69]. The keywords nestled at the centre of the map has the strongest TLS, and highest co-occurrence with many other terms, in contrast to the terms at the edge of the map, occurring with smaller number of keywords. The distance between each keyword identifies as the strength of relation of association. A shorter distance reveals a stronger relation.

In Figure 2A, five clusters can be identified, represented by colours red, blue, yellow, purple and green. Clusters represents the division of keywords into subject area or research theme [73]. The first cluster in yellow is classified as the soil pollution, while the second cluster in purple is identified as water pollution. Unsurprisingly, water and soil are the major victims of hydrocarbon pollution since petroleum oils are often spilled into seawater or soil [4,5,6,7]. An oil spill on land contaminates the soil and subsequently allows the dissemination of oil pollutants from soil into groundwater. Similarly, the spilled oils present in seawater are transferred to the coastline by water current, thereby contaminating the marine soil and sediment. Although both soil and water are closely related to hydrocarbon pollution, the clean-up treatments at polluted sites differ significantly from each other. Therefore, two major research themes revolving around soil and water were identified and grouped respectively, suggesting efforts made to study and tackle these pollutions in cold regions. The word “bioremediation” is most mentioned across all 32 analysed documents and closely linked to “biodegradation”, “soil pollution” and “soil pollutant”. Many theories can be proposed on the feasibility of bioremediation of hydrocarbons for soil pollution, however, this is probably pointing to the sure fact of biodegradation of oil happening among soil degraders, naturally occurring in the environment. This piece of information has spurred many studies over the years, which has finally led to the development of technologies that can harness this into something beneficial for mankind and the environment.

Another cluster in green highlights the biodegradation of hydrocarbon. Within the green cluster, keywords of “biodegradation” and “temperature” ranked the top and second highest for TLS, suggesting a strong correlation between them. Undoubtedly, temperature plays an important role in stimulating the hydrocarbon biodegradation rate at its natural environment. Throughout the years, the optimisation of the temperature parameter has been studied and reported to determine the optimal temperature that favours the microbial growth in response to the extreme conditions. A large amount of hydrocarbon-degrading microbes, theoretically, generates a higher removal capacity, resulting in an increase of biodegradation efficiency.

The fourth cluster in red depicts studies revolving around molecular and microbiology. The keyword with the highest number of occurrences in this cluster is “environmental” with a value of 122 occurrences. The keyword “environmental” is closely linked to “microbiology”, “phylogeny” and “classification”. Microbial bioremediation has been proven effective in remedying the petroleum-polluted soils. The reduction of these pollutants is contributed by the presence of hydrocarbon-degrading bacteria in the environment. The biological analysis of native microorganisms has received extensive attention due to strict laws in cold regions that forbid the use of foreign microorganisms. Hence, to better understand these bacterial groups, environmental samples from its native place have been collected and investigated over the years. A detailed analysis on the hydrocarbon-degrading bacteria provides a groundwork for the efficient implementation of bioaugmented-bioremediation to remove hydrocarbon pollutants.

The last blue cluster suggests surface-active bio-substance studies, in which the word “crude oil” is closely linked to “bacteria” and “surfactant”. In cold regions, the low temperature reduces the bioavailability of oil pollutants for biodegradation. To tackle this, biosurfactant agents have been proposed to hydroxylate these recalcitrant pollutants into a readily biodegradable form. This data has driven many studies that explore and exploit native biosurfactant-producing bacteria into a beneficial tool for the clean-up treatment. Figure 2B is a distribution of keywords according to average publications year using overlay visualisation generated from VOS viewer software. Keywords that appeared in early years are coloured in blue and yellow in later years. Here, readers can deduce the direction and interest of more publications across years. The words “petroleum pollution” and “biosurfactant” seem to be the current research focus to elucidate the actual role played by biosurfactant-producing microbes. The addition of biosurfactant-producing microbes into bioremediation can potentially stimulate the bioavailability of pollutants in response to cold temperature. Detailed analysis of the biosurfactant effects on hydrocarbon biodegradation enhances the understanding of biosurfactant-assisted bioremediation to treat petroleum pollution.

## 3. Enhanced Bioremediation Studies in Cold Regions

In general, bioremediation can be categorised into two different groups, subjecting to the different implementation strategies and availabilities. The first group is the in situ bioremediation, in which the removal treatment is performed at its native location without any geographical transfers, while the latter group is the ex situ bioremediation, which involves the translocation of contaminated substances from its native place into a treatment plant [23]. Both bioremediation approaches are beneficial to mitigate petroleum hydrocarbon pollutants in cold environments (Figure 3), hence their pros and cons have been summarised in Table 3.

### 3.1. In Situ Applications

In situ bioremediation is an on-site treatment that utilises the native biogeochemical response within the contaminated site to eradicate petroleum pollutants without excavation and translocation. This treatment generates no physical effects on soil structures. Under cold regions, in situ bioremediation has been applied to address some constraints, including costly soil excavation that prohibits ex situ implementation, deeper soil pollution that restricts the efficiency of other removal methods and the conservation of essential soil structure near polluted sites [85]. However, in situ bioremediation is relatively uncontrollable and naturally less effective. Thus, biotechnological improvements were made to tackle these limitations, including soil aeration, heat supply, microbial manipulation and cocultivation. Examples of enhanced in situ bioremediation under cold conditions are phytoremediation, bioventing and biosparging (Table 4). Natural attenuation is a natural process; thus, it will not be discussed in this review.

#### 3.1.1. Phytoremediation

Hydrocarbon phytoremediation is defined as the on-site exploitation of indigenous, psychrotolerant plants to eradicate petroleum pollutants found in soils, sediments and water bodies [86]. This phytoremediation can utilise physical, chemical or biological mechanisms depending on the contaminant’s nature. The success of phytoremediation in cold conditions is affected by many parameters, such as the psychrotolerant plant rooting system in response to pollutant depth (fibrous- or tap-rooted plant), the pollutant type (relatively degradable linear hydrocarbons or highly persistent aromatic hydrocarbons), the plant survival rates (toxicity of pollutants) and most importantly, the time period for a complete clean-up [87]. Thus, different phytoremediation approaches have been proposed to eradicate different kinds of pollutants, including phytodegradation, rhizodegradation, phytovolatilisation, phytoextraction, rhizofiltration and phytostabilisation [88]. Among these methods, phytodegradation and rhizodegradation are predominantly observed under cold conditions [16,85,89]. Phytodegradation is a process in which hydrocarbon pollutants are absorbed, stored and broken down in the plant tissues. Meanwhile, rhizodegradation disintegrates hydrocarbons owing to the mutualistic interaction between soil microbiomes and plants [85,90].

To date, many native phytoremediators such as trees, shrubs and grasses have been exploited to mitigate hydrocarbons present in contaminated soil under cold conditions [43,75,76,91]. The cocultivation of different plants within a treatment site is proven feasible to achieve a satisfactory clean-up by catalysing the phytodegradation process. This approach is widely applied in Arctic regions that possess various plant species that can degrade hydrocarbons and survive at low temperature. A field study in Alaska reported a significant hydrocarbons reduction by 95% after 15 years of treatment on polluted soil with an initial contaminant concentration of 8300 mg/kg [75]. The satisfactorily clean-up of hydrocarbons by *Salix bebbiana, Salix alexensis, Salix glauca, Betula neoalaskana, Picea glauca* and *Populus balsamifera* was stimulated by the hospitable environment at the treatment site [75]. This hospitable environment was achieved by adding high organic contents at the initial stage (fertilisers added to support primary planting) and attaining higher water retention (cocultivation of different plants). The cocultivation of native woody plants and native grasses enhances the TPH biodegradation, in which the former group has deeper roots to render hydrocarbons dissemination for the long-term treatment, while the latter group has surface roots for initial petroleum removal. A pilot field study in Sweden further supported the cocultivation method in phytoremediation treatment by showing a significant reduction of polycyclic aromatic hydrocarbons (PAHs) in polluted soil from a combination of sunflower (*Helianthus annuus*) and alfalfa (*Medicago sativa*) [43]. A similar was finding also reported by Palmroth et al. [76] in petroleum-polluted subarctic soil using cocultivation of pine trees, grasses and legume plants.

On the other hand, hydrocarbon rhizodegradation by native plants and microbial endophytes is another removal method in cold regions. Endophytic microorganisms are defined as any mutualistic microbes that reside on the host plant organs and tissues, developing a beneficial relationship between them [91]. For instance, the host plant serves as a nutrient contributor by supplying its essential carbohydrates to endophytic bacteria, while these bacteria facilitate the host plant to grow and survive in cold environments by reducing the biotic and abiotic stresses. A study by Ferrera-Rodríguez et al. [91] highlighted the crucial role of endophytic bacteria (i.e., Actinobacteria) with a host plant (*Puccinellia angustata*) in removing petroleum contaminants under Arctic environment. In the study, a high hydrocarbon degradation rate was recorded with psychrotolerant *Arthrobacter* sp., *Rhodococcus* sp. and *Sanguibacter* sp. found in the rhizosphere of *P. angustata*. Further analysis using polymerase chain reaction (PCR) revealed the upregulation of three nalkane degradation genes (i.e., alkane hydroxylase, alkB, naphthalene dioxygenase, ndoB, and 2–3-catechol dioxygenase, xylE) within these endophytic bacteria, leading to a temporal rhizospheric effect in petroleum contamination incidents [91]. In other words, the nutrient provided from host plants stimulates the growth of bacteria and upregulates the hydrocarbon-degrading genes, resulting in a significant reduction of hydrocarbon pollutants.

Overall, this green technology has many benefits, such as cost-effective (no liability charges from sophisticated equipment, transportations and waste disposals), aesthetic enhancement, prolonged treatment without any maintenance requirement, soil structure conversation and harmless cleaning process that relies on natural solar energy [75,85,92,93]. However, in some circumstances, the animal food web could be compromised when the inhabited animal consumes these phytoremediators; hence, non-targeting food plants are highly preferred in phytoremediation treatments. In Antarctica, phytoremediation is less effective to treat polluted soils due to limited plant availability (abundant in mosses and lichens) that can degrade hydrocarbons and grow under extreme cold temperatures. Nevertheless, co-cultivated phytodegradation and mutualistic rhizodegradation are feasible on-site removal methods to mitigate hydrocarbons in other cold regions such as the Arctic and Alaska, perceiving that they are natural cleaning processes, and no on-site supervision is needed to sustain the ongoing removal treatment.

#### 3.1.2. Bioventing and Biosparging

Bioventing in cold conditions supplies oxygen via air injection for indigenous, cold-adapted microorganisms to promote the microbial biodegradation. The enhanced bioventing is carried out at its native contaminated site where the oxygen is supplied via an air vessel that is installed into the subsurface soil. This treatment degrades volatile pollutants into relatively non-toxic vapours that propagate slowly through soils [46]. Influencing factors of bioventing have been identified, such as the soil permeability, humidity, presence of interfering compounds and oxygen availability [85]. The soil permeability and humidity control the petroleum degradation by regulating vapour propagation, while the existence of air disturbance compounds disrupts the air injection fluency, and lastly, the oxygen availability regulates the aerobic biodegradation. These constraints can be overcome by adding nutrients and manipulating the soil moisture or applying ozonation [78,94]. Nutrients are essential energy for microbes to grow and degrade hydrocarbons, while ozonation is used to accelerate partial oxidation of recalcitrant pollutants into volatile vapours. Enhanced bioventing has been proven effective for the clean-up of the small-scale, medium molecular weight hydrocarbon pollution in soils owing to the higher oxygen supply [95]. Medium-weight hydrocarbons such as diesel, bitumen and gasoline are volatile compounds that can volatilise into smaller BTEX (i.e., benzene, toluene, ethylbenzene and xylenes) vapours [95]. Besides petroleum hydrocarbons, bioventing is also used in anaerobic bioremediation of chlorinated pollutants. A study demonstrated that the replacement of oxygen air with the mixture of nitrogen and a low amount of carbon dioxide (electron donor) generates a significant removal of recalcitrant chlorinated pollutants (i.e., 1,1,1-trichloro-2,2-bis(p-chlorophenyl)ethane, DDT, and 2,4-dinitrotoluene, DNT) in soil [96].

Similarly, biosparging utilises oxygen injection in the treatment at its native site. The air is injected into saturated soils, leading to the propagation of volatile contaminants from its native place to the soil surface that possess hydrocarbon-degrading microorganisms [97]. Saturated soil is soil that contains little or no pores, while unsaturated soil is soil that has many pores. The purpose of this air injection is to transfer volatile pollutants from a deeper depth to the soil surface, where higher oxygen levels and presence of hydrocarbon-degrading microbes are observed. Two concerning factors are reported to influence biosparging treatment, including the soil permeability and the recalcitrance level of pollutants [94]. A highly permeable soil favours the transfer of contaminants, while highly degradable pollutants generate higher removal efficiency. According to a study by Wu et al. [98], a significant removal of benzene pollutants (96%) has been observed in biosparging treatment on soils with deeper depth. Another biosparging study by Kao et al. [99] reported the removal of 70% BTEX contaminants from the BTEX-polluted groundwater. These studies suggest biosparging can be used in deeper soil and groundwater containing hydrocarbons pollutants.

In cold regions, limited bioventing and biosparging treatments have been reported due to the low temperatures that freeze the soil (no oxygen) and pose challenges to the treatment. This phenomenon restricts hydrocarbon removal, resulting in lots of untreated sites that stay for years. To tackle these limitations, the delivery of air via injection has been designed and optimised to ensure fluent and sufficient oxygen supply. For instance, a microbioventing that contains nine tiny air injection tubes of a size 0.5 m was designed and optimised by Rayner et al. [46] to remediate polluted soil in sub-Arctic regions. The nine optimised injection points stimulate uniform air distribution on the surface of TPH pollutants, thus increasing the exposure of oxygen to hydrocarbons-degrading microorganisms. By using microbioventing, a significant TPH biodegradation rate of 1020 mg kg^−1^ per day was reported and the complete clean-up of an initial TPH concentration of 7000 mg kg^−1^ can be achieved within a two-year treatment [46]. In response to the acclimated soil containing TPHs under cold conditions, King et al. [77] optimised bioventing treatment on aged soil (approximately 2 to 3 years). From the study, bioventing with an airflow speed of 275 cm^3^ min^−1^ was found effective on aged soil, with a significant 82% removal under a treatment that mimics the summer temperature of 10 °C. The findings suggest that bioventing with an optimised airflow rate can be applied to remediate hydrocarbons in aged soils effectively.

### 3.2. Ex Situ Implementations

Ex situ bioremediation requires an additional excavation and transportation of contaminated soils to a specific treatment site equipped with infrastructure and guaranteed safety measures. A well-established treatment site can significantly enhance the petroleum remediation efficiency by offering the most optimum environmental requirements for the treatment [38]. The high preferences of ex situ bioremediation in cold environments are attributable to the stable and higher remediation efficiency throughout the large-scale treatment [38,100]. Examples of enhanced ex situ bioremediation in cold regions are biopile and landfarming (Table 5). A potential ex situ bioreactor approach to treat polluted soils is also presented.

#### 3.2.1. Biopile

Biopile is proven effective and feasible to remediate hydrocarbon-polluted soils. In cold regions, biopile-facilitated bioremediation utilises an elevated piling of the contaminated soil excavated from a pollution area. To increase the hydrocarbon removal efficiency, irrigation, organic fertiliser and treatment pile are applied in the biopile treatment in response to the cold condition. Some benefits of biopile have been proposed, such as space-saving by elevated pile and high removal efficiency provided that the fertiliser bioavailability, soil temperature and aeration are satisfactorily supported [15,44,48,49].

Temperature plays an important role in biopile treatment to clean-up hydrocarbon pollutants under cold conditions. An optimum soil temperature (5–10 °C) with nutrient amendments can generate a favourable environment for microbial biodegradation and regulate the volatilisation of low-molecular weight hydrocarbons [15,48,49]. Recently, a half-tonne biopile study in Antarctica highlighted the effect of total sunlight exposure time on its removal efficiency [49]. From the study, higher hydrocarbon removal (75%) was observed in biopile with 157 h of total sunlight exposure compared to the second biopile (55%) with a total of 108 h. The findings suggested that higher sunlight exposure can increase the soil temperature and favour microorganisms’ growth as well as hydrocarbons’ biodegradation. Another temperature-dependent biopile study on TPH in Antarctica also reported a significant reduction of 75% in summer compared to other seasons, which generally has a higher temperature that enhances microbial activity [48]. Álvarez et al. [15] reported that a 12-month biopile with a mean temperature of 6.5–6.7 °C displays a superior hydrocarbon removal efficiency (75.8%) compared to the control (49.5%), with a mean temperature of 5.2–5.3 °C. All the above studies applied fertiliser to stimulate the growth of hydrocarbon-degrading microbes for efficient biodegradation. These data revealed the significant effects of temperature on biopile under cold conditions to clean-up petroleum hydrocarbons when sufficient nutrients are provided. However, the temperature requirement should be optimised prior to the actual implementation since the excessive heat can kill and inhibit psychrotolerant microbes, as suggested by Sanscartier et al. [101].

Bioaugmentation has been applied in biopile treatment to treat polluted soils. A bioaugmented biopile study by Gomez and Sartaj [44] using mature municipal compost and bacterial consortium reported a higher TPH removal of 82% compared to the control biopile (48%). The bioaugmented biopile is an enhanced bioremediation supplied with hydrocarbon-degrading microbes and nutrient amendments. The study suggests the significant remediation effects of adding oil-degrading microbes to stimulate microbial biodegradation of hydrocarbons, resulting in a higher removal efficiency. In other words, theoretically, the more hydrocarbon-degrading microorganisms in a treatment, the higher the removal ability. Interestingly, a study in Ireland by Germaine et al. [79] proposed a feasible phytoremediation-mediated biopile (known as “Ecopile”) treatment in hydrocarbon-contaminated soil under cold regions. From the study, a nearly complete clean-up of hydrocarbon pollutants (lower than detectable thresholds) was achieved after one-year treatment using two phytoremediators (perennial ryegrass, *Lolium perenne* and white clover, *Trifolium repens*) inside the biopile treatment site. The findings supported the theory of having more oil-degrading players that eventually generate higher hydrocarbon remediation efficiency. Cost-saving is highly attainable by implementing the “Ecopile” method due to the low labour and maintenance fees needed for the small-scale treatment; yet, more scientific data such as toxicity effect of hydrocarbons and biodegradation sustainability of phytoremediator are needed before the actual implementation of “Ecopile” in the future.

#### 3.2.2. Landfarming

Polyaromatic hydrocarbon-polluted soils retrieved from a soil depth of more than 1.7 m hinder the microbial biodegradation due to low oxygen availability [102]. Thus, landfarming in cold environments excavates these soils into a treatment bed by spreading them evenly on its surface [47]. This treatment is conducted at a different location, where irrigation devices are provided. Benefits of landfarming bioremediation have been highlighted, such as no extensive pre-assessment on contaminated sites prior to the treatment, and large treatment capacity with minimal supervision needed [93]. Under cold conditions, landfarming is hindered by some limitations that include freeze–thaw cycles and soil humidity [47]. Thus, nutrient addition and soil irrigation or tillage (soil mixing to supply air and thaw–freeze soils) have been proposed as their improvements.

An enhanced landfarming is referred to as landfarming that has been improved by fertilisers amendment and irrigation. According to Paudyn et al. [47], a three-year field study in the Canadian Arctic reported a significant TPH removal efficiency of 80% in diesel-polluted soils by implementing enhanced landfarming with rototilling techniques. The rototilling has been optimised for landfarming by turning over the polluted soil and at the same time, spraying nutrients into soils. This practice allows sufficient oxygen supply and enough nutrients to stimulate growth of microorganisms. Similarly, another study in the Canadian Arctic reported a high reduction of TPH by 60% after a two-month enhanced landfarming treatment with a periodic ten-day tilling [45]. Noteworthily, McCarthy et al. [30] reported that a total clean-up of BTEX and gasoline-range organic compounds in contaminated soil was achieved within two months of enhanced landfarming bioremediation (nutrient amendment and aggressive tilling) in the Southwest Barrow, Arctic. The aggressive tilling was used to render the BTEX volatilisation, making them available for aerobic biodegradation. Overall, these studies demonstrated a significant combined effect of nutrient additions and tillage techniques in landfarming treatment on TPH removal (more than 60% reduction within a shorter time period) under cold conditions.

However, a recent plot study in Italy demonstrated different findings by comparing the TPH removal efficiency between bioaugmented landfarming, enhanced landfarming and natural attenuation after 90-day treatment [80]. Bioaugmented landfarming is referred to as enhanced landfarming that receives an addition of beneficial hydrocarbon-degrading microbes into the treatment site supported with nutrients and tillage system. From the study, bioaugmented landfarming was recorded with the highest average TPH removal percentage of 86%, as compared to the enhanced landfarming (70%) and natural attenuation (57%), respectively. The significant TPH reduction in bioaugmented landfarming was contributed by the cohabitation of autochthonous microorganisms in the treatment site. These bacteria have a wider range of hydrocarbons biodegradation capability by using diverse catabolic enzymes possessed by different degrading strains [80].

Interestingly, a study by Jeong et al. [42] proposed a bioaugmented landfarming without the use of a tillage technique to stimulate the microbial biodegradation. This bioaugmented landfarming applies surfactant foams containing native, hydrocarbon-degrading bacteria and nutrients that are sprayed twice per day on hydrocarbon-polluted soils. From the study, the foam-sprayed landfarming (no tillage) displayed a superior TPH removal of 73.7% compared to the enhanced landfarming (46.3% with tillage) at 6 °C. The higher TPH removal in foam-sprayed landfarming was due to the aqueous foam that moisturise oil from polluted soils into a readily biodegradable aqueous form [42]. Since tillage systems are not applied in this treatment, the operational cost is highly reduced while the efficient hydrocarbon removal ability is conserved.

Studies have reported that improper addition of bioaugmented microbes to the treatment can be useless and a waste of resources. Therefore, more clarifications on the bioaugmented landfarming (nutrient + irrigation + hydrocarbon-degrading microbes) are needed in Antarctica, the Arctic and other cold regions to compare the TPH removal efficiency with enhanced landfarming (nutrient + irrigation). Information such as suitable native microbes that possess high removal ability and grow well in genuine cold environments is highly preferred. Since bioaugmentation is influenced by biotic and abiotic factors, the foam-spraying technique is a possible way to introduce them into a treatment site. But the viability study (how long the microbes can survive in surfactant foams) of bioaugmented microbes on the surfactant foam has not been investigated, hence the role of these foams is uncertain.

#### 3.2.3. Bioreactor

Bioreactor is a bioremediation technique that utilises a vessel to transform toxic pollutants into less toxic, smaller compounds using a biological reactions cycle [81]. Different categories of bioreactor operational mode including batch-based, semi-continuous-based, continuous-based and multistage-based treatment can be chosen depending on the expenditure budget and market financial availability [16]. Bioreactors can be generally manipulated into compost-based and slurry-based, where the slurry-based approach has the capability of eradicating petroleum hydrocarbon pollutants persisted in soil [103,104]. The slurry-based bioreactor is conducted by adding excavated polluted soil with water that produces slurry products in a vessel supported with the continuous agitation, specific nutrient amendments and oxygen supply [81,103,104]. The major benefit of bioreactor treatment is the optimal operational setting (pH, humidity, temperature and nutrient bioavailability) that favours TPH biodegradation, giving a higher removal rate than other ex situ bioremediations [85,105]. This bioreactor mimics and conserves natural environmental conditions that allow maximum growth of microbial activity, thus TPH concentration in soil can be significantly reduced. The flexible bioreactor experimental design also reduces abiotic losses (enclosed treatment space) and allows microbial population characterisation study (data on microbial shifts after short- or long-term study) [82,83].

The confined bioreactor system also enables the utilisation of genetically modified microorganisms (GMMs) that favour TPH biodegradation, and these GMMs can be destroyed after usage to conserve the integrity of the native environment [16]. This practice provides a possible use of GMMs in TPH-contaminated soil in Antarctica and other extreme cold regions since most of these countries restricted the usage of GMMs in bioremediation treatment. However, bioreactor treatment on contaminated soils is usually time-consuming due to the tedious experimental design (parameter optimisation using one-factor-at-a-time, OFAT), high cost attributable to the higher manpower number and transportation budget (especially on large-scale polluted soil) and requiring additional post-operational procedures (i.e., soil desiccation and wastewater treatment) [83,84]. Therefore, this bioreactor technique is rare for the full-scale practice under cold conditions.

To date, bioreactors are widely applied in the wastewater treatment under the polar Arctic Circle to eradicate nitrogen and phosphorus contaminants persisted in effluents [106,107]. For instance, an enhanced bioreactor supplemented with a mixed population of bacteria, *Archaea* and fungi demonstrated an efficient removal of organic matter (>96%), ammonium (>95%) and nitrogen (75%) in polluted effluent at an average temperature of 10 °C. Although there is limited real-field bioreactor treatments that reported on the removal of TPH in contaminated soils under cold regions, preliminary studies have proposed a feasible removal by adding psychrotolerant, hydrocarbon-degradative bacteria into the slurry-based bioreactor system to remove these pollutants [51,54,108]. Thus, more scientific exploration on the real-field bioreactor treatment to treat hydrocarbon-polluted soils is greatly needed to elucidate the applicability and sustainability of the bioreactor approach under cold regions.

## 4. Other Potential Applications

Besides biostimulation and bioaugmentation, molecular and microbial applications can be exploited to stimulate hydrocarbon remediation efficiency. With molecular techniques, GMMs and engineered nanozymes are studied and constructed to improve the removal ability. Meanwhile, recent immobilised cell systems and microbial biosurfactant have been reported beneficial to increase microbial stability and activity. Thus, these potential techniques can be applied in response to the unfavourable environmental condition in cold regions.

### 4.1. Genetic Engineering

The recent advancement in genetic engineering potentially allows effective bioremediation of hydrocarbon pollutants using artificial bacteria consortium or GMMs. Recombinant GMMs are prepared by cloning the beneficial genetic materials into cold-adapted microbes. The inoculated genetic materials can be in any forms, including a single gene cluster of desirable catabolic pathway and a modified prevailing degradative gene [109]. The recombinant technology allows the overexpression of targeted genes into larger amounts while maintaining its stability and activity [109]. These stable GMMs are higher in quantity that produce aggressive catabolic activity, resulting an increased removal efficiency.

This technique is potentially useful to transfer prevailing degradative genes from a specific environment to other cold environments, targeting degradation of specific pollutants. A study by Luz et al. [110] reported that biphenyl dioxygenase (*bphA*) and toluene dioxygenase (*todC1*) were the most prevalent hydrocarbon-degradative genes in Antarctica. These genes are essential aromatic dioxygenases that catalyse the oxidation of aromatic compounds such as naphthalene, salicylate and toluene [110]. Thus, potential GMMs can be produced by inoculating these dioxygenase genes to degrade aromatic hydrocarbon pollutants at low temperatures. A psychrotolerant recombinant strain, *Pseudomonas* sp. Cam-10, was designed to biodegrade polychlorinated biphenyl (PCBs) contaminants (i.e., naphthalene, salicylate, 2-chlorobiphenyl and 4-chlorobiphenyl) at a lower temperature of 7 °C [111]. The ability to degrade PCBs successfully at low temperature by strain Cam-10 was due to the inoculation of the *Bph* gene cluster (encoding biphenyl dioxygenase) and the optimisation of *lacZ* reporter. There are limited cold-adapted recombinant strains reported in hydrocarbon bioremediation in response to the cold condition (Table 6).

The pioneer study using psychotropic *Pseudomonas putida* Q5T revealed the feasibility of recombinant technology to eradicate specific environmental pollutants under cold temperatures [112]. From the study, a recombinant *P. putida* Q5T was constructed to degrade toluene successfully at 0 °C after the inoculation of TOL plasmid isolated from the mesophile, *P. putida* PaWl [112]. The findings served as a good example to demonstrate the exploitation of cold-adapted, recombinant GMMs in bioremediation treatment. Another recombinant Antarctic strain, *Pseudoalteromonas haloplanktis* TAC125, was constructed with the toluene-o-xylene monooxygenase gene (i.e., *ToMO*) isolated from mesophilic *Pseudomonas stutzeri* OX1 [113]. The inoculation of the *ToMO* gene allowed the recombinant TAC125 to effectively degrade various hydrocarbons such as benzene, phenol, xylene, toluene and naphthalene at 15 °C. The toxic catechol by-products generated after the hydrocarbon biodegradation can be removed by the prevailed *PhcopA* gene (oxidative removal by periplasmic putative laccase-like proteins) present in the recombinant strain, suggesting higher sustainability [113]. Meanwhile, contaminant of 2,4-dinitrotoluene (DNT) can be removed by *Pseudomonas fluorescens* RE at 10 °C after the insertion of the *dnt* gene cluster (2,4-DNT dioxygenase and 4-methyl-5-nitrocathecol monooxygenase) retrieved from the *Burkholderia* sp. strain DNT [114]. These data revealed and proposed the feasibility of GMMs in hydrocarbon removal treatment under cold conditions.

Interestingly, a recent transgenic tobacco plant, *Nicotiana tabacum,* was reported using the bacterial *Ntr* gene (nitroreductase) to degrade DNT pollutants at 4 °C [115]. The objective of the study was to produce a transgenic plant used in the phytoremediation treatment to remediate DNT pollutants during the winter season. The overexpression of beneficial cold-adapted genes in phytoremediators may be a feasible method for removing toxic pollutants under cold regions. However, data on toxicity study and removal time evaluation are greatly needed to assess the sustainability of these transgenic plants in cold environments. Despite these beneficial results, cold-adapted GMMs are underdeveloped and seldom applied in the real-field treatment due to the prohibition of biosafety concerns and strict environmental regulatory laws in certain countries such as Norway, Sweden, Iceland and Antarctica [23]. However, the confined bioreactor provided a potential use of these GMMs to remove hydrocarbon pollutants in polluted soils since the foreign genes can be destroyed after usage while the integrity of the environment is conserved.

### 4.2. Enzyme Engineering

Potential enzyme engineering can be utilised with genetic engineering to improve the hydrocarbon bioremediation efficiency. Enzyme engineering is a molecular alteration of the amino acid structure in a targeted enzyme to enhance its catabolic activity. An engineered enzyme can tolerate environmental stresses with higher structural stability and stimulate substrate specificity [116]. The structural stability and functionality of the enzyme is highly dependent on the amino acid sequence; thus, the specific alteration using recombinant DNA can be performed to generate the desired active enzyme [117]. For instance, the enhanced biodegradation of PAHs (naphthalene and anthracene) has been demonstrated by the mutant cytochrome P450 BM-3 isolated from mesophilic bacteria, *Bacillus megaterium* [118]. The efficient removal of these PAH compounds was due to the site-directed mutagenesis of three functional amino acids (Phe87, Leu188 and Ala74), improving the substrate binding efficiency to the active site coupled with minimal NADPH consumption [118]. Also, mutant BM-3 has been reported to hydroxylate recalcitrant four-ring compounds such as pyrene and chrysene, thus increasing the bioavailability of these water-soluble hydroxylated compounds for the biodegradation to take place [119]. In other words, enzyme engineering can stimulate hydrocarbon removal efficiency, as shown in the engineered BM-3 enzyme under tropical temperatures. In cold regions, enzyme engineering has not been studied and applied in hydrocarbon bioremediation, yet previous data from tropical regions suggest that structural alteration on degrative enzyme can improve the removal efficiency. Thus, more studies are greatly needed to design and evaluate the structural manipulation of cold-adapted enzymes.

Enzyme engineering can also be modified with nanotechnology to produce nanozymes for the bioremediation of environmental pollutants. Nanozymes are defined as the next-generation synthetic enzymes with a size smaller than 100 nm that mimic enzyme-like characteristics [116]. These nanozymes have many benefits, such as low production cost, high robustness and high enzymatic stability. Different nanomaterials have been proposed in mimicking many natural enzymes such as nanocrystals, nanotubes, nano-sponges, nanoparticles, nanomembranes and nanocomposites to eradicate various environmental pollutants [120]. It has been reported that synthetic enzymes incorporated within nanomaterials are capable of catalysing the substrate transformation using the exact kinetic pathway developed in the natural enzyme (i.e., catalase, oxidase and peroxidase), thus raising scientific attention in the nano-bioremediation [120]. Bioremediation that applies nanozymes to treat pollutants is referred to as nano-bioremediation.

Studies have reported that nano-bioremediation can successfully remove toxic contaminants from the environment. For instance, toxic polybrominated diphenyl ethers have been successfully degraded by 67% using nanozyme of nZVI (nanoscale zero-valent iron) and a hydrocarbon degrader of *Sphingomonas* sp. PH-07 [121]. The study highlighted that the combination of nZVI and strain PH-07 had adapted effectively to a higher contaminant concentration of up to 5 g/L, with a higher removal efficiency compared to the conventional study [121]. Another recent successful hybrid phytoremediation with the addition of bimetallic nanoparticles (palladium and iron) and humic acids revealed a significant removal of hexabromocyclododecane by 99% in aqueous and 27% in soil, respectively [122]. Although there was a major difference in the removal efficiency between soil- and water-based studies, both treatments were reported with higher degradation rate as compared to their respective control. The study also suggested the humic acid associated with bimetallic nanoparticles as an excellent enhancer to remove organic pollutants persisted in soils by facilitating efficient electron transfer for the degradation response [122]. These hybrid bioremediation with nanozymes demonstrated satisfactory findings; however, such scientific claims have not been studied and reported in cold regions. Hence, nano-bioremediation can serve as a potential removal approach for the hydrocarbon-polluted soils in cold environments, while more scientific data are essential to better understand the removal efficiency and toxicity of nanozymes under low temperatures.

### 4.3. Immobilisation Tools

In response to the enhanced bioremediation via bioaugmentation, suitable approaches to introduce hydrocarbon-degrading bacteria or microbial consortia into the treatment site are greatly needed under cold environments. Immobilisation tools have been highlighted as promising techniques to inoculate bioaugmented bacteria into the treatment site for the removal of pollutants [123,124]. Immobilisation applications restricted the movement of integrated cells or functional degrading enzymes within a specific matrix and allowed them to mimic the ability of biofilm production on contaminants’ surfaces seen in naturally occurring bacteria [123,124].

Former bioremediation treatments in cold regions applied the free cell culture to remove hydrocarbons present in polluted soils, yet the limited stability and activity of these free-cell bacteria were reported [125]. On the other hand, the immobilised inoculum improves the sustainability and functional catalytic ability of these cells and enzymes, allowing long-term usage to reduce bioremediation cost [23]. To tackle these limitations, studies have been conducted to evaluate the removal efficiency and sustainability of immobilised cultures on the removal of pollutants at low temperature. A cold-adapted *Pseudomonas plecoglossicida* strain TA3 was entrapped into an agar cube to remove carbamate pollutants [126]. An agar cube was selected as an entrapment media owing to a few benefits, such as inexpensive, higher biomass and higher cell stability. In comparison, the immobilised TA3 cells had the higher carbamate removal efficiency of 80% compared to the free cells of TA3 (60%) at 4 °C [126]. The enhanced biodegradation of carbamate by immobilised TA3 cells was contributed by beneficial factors such as greater mechanical strength within agar matrix, higher cell density and better stress tolerance, as suggested by the study [126].

Another research used a polyurethane foam matrix to encapsulate two oil-degrading strains (*Pseudomonas* monteilii P26 and *Gordonia* sp. H19) for the removal of petroleum oil pollutants in polluted sea water [127]. Polyurethane foam generally has superior mechanical characteristic, higher porosity and greater adsorption ability that generate better microbial stability [128]. The study compared the removal efficiency of petroleum oil between the fresh- and aged-immobilised cells at 4 °C, in which the latter group displayed a higher degradation (55.5%) than the former group (33.5%). The study reported that the enhanced degradation ability in low temperatures by aged-immobilised cells was attributable to the mixed biofilm production and bacteria acclimation that improve their metabolic activity [128]. Immobilised *Pseudomonas* sp. DJ1 with peat and sawdust matrix also displayed a higher oil degradation of 77.3% compared to the non-immobilised system (70.6%) at 5 °C [129]. These studies proposed the repetitive use of immobilised cultures in the treatment coupling with longer shelf-life to substantially reduce the operational cost.

Nevertheless, there are limited studies reported on enhanced biodegradation via immobilisation tools in low temperatures. A review has summarised the implementation of immobilisation tools using cold-adapted enzymes in potential industrial and environmental applications, suggesting a possible use of this technique in response to cold conditions [125]. The large fluctuation of temperatures in cold regions restricts the microbial activity; hence, immobilised techniques can be potentially used to get rid of such constraint. According to Lee et al. [125], suitable selective carriers and appropriate immobilisation techniques differ for each sample based on environmental settings and cellular properties. Thus, information on suitable carriers and hydrocarbon removal efficiency in low temperatures are crucial to evaluate and apply immobilised cell systems to treat polluted soils in cold regions.

### 4.4. Microbial Biosurfactant

In general, biosurfactants improve hydrocarbon bioremediation efficiency by inoculating biosurfactant-producing microorganisms into the contaminated soil. Microbial biosurfactant is defined as an amphiphilic molecule produced by microorganisms to promote bioavailability of pollutants for the microbial biodegradation [130]. In cold regions, the limited bioavailability of petroleum hydrocarbon pollutants generates a lower removal rate owing to the recalcitrance oil pollutants that resist biodegradation [131,132]. To tackle this limitation, microbial biosurfactants are applied to emulsify hydrocarbon compounds and subsequently promote the uptake of these pollutants for biodegradation. Many cold-adapted bacteria have been reported to generate biosurfactant molecules upon the conditional stress from the harsh environment. These psychrophilic bacteria genera include *Rhodococcus*, *Pseudomonas*, *Pseudoalteromonas*, *Idiomarina* and *Pantoea* that can degrade hydrocarbons and produce biosurfactants [131,132,133,134]. These biosurfactants are environmental-friendly, stable and easily biodegradable, thus gaining increased interest to apply in hydrocarbon bioremediation treatments.

The stable biosurfactant allows an effective hydrocarbon biodegradation regardless of the effects of extremely low temperature, pH and salinity, as shown in a study by Xia et al. [135]. The study reported a highly stable rhamnolipid JBR-425 produced by *Rhodococcus erythropolis* OSDS1 that can tolerate a wide range of pH (4–9), temperatures (4–100 °C) and salinities (0–100 g L^−1^ sodium chloride concentrations). The addition of this emulsifier, *R. erythropolis* OSDS1, produced another 10% degradation improvement to achieve a final degradation efficiency of 85%, suggesting the enhanced bioavailability of crude oil from rhamnolipid JBR-425. The findings of Luong et al. [132] also reported that trehalolipid biosurfactants produced by *Rhodococcus erythropolis* S67 can facilitate the uptake of hexadecane for microbial biodegradation at low temperatures. In the study, the culture surface tension was reduced from 70 to 45 mN m^−1^ with an emulsification index of 8% after 12 days, suggesting the formation of micelles on hydrophobic substrates. A micelle is a highly soluble amphiphilic molecule with its hydrophilic head facing the aqueous culture medium. The micelles enhanced the solubility of hexadecane for biodegradation, resulting in a substantial removal of 30% at 10 °C after 12 days [132].

Biosurfactants can be a promising technique to enhance the hydrocarbon bioremediation efficiency in cold environments. However, most biosurfactant studies were conducted on a laboratory-scale and a successful case has not been reported on a real-field implementation under cold regions. Thus, the real-field bioremediation treatments with microbial biosurfactants are necessary to assess the hydrocarbon removal efficiency under genuine environmental conditions. The assessment of toxicity effect of biosurfactants on the ecosystem also needed to further evaluate the suitability of these microbial biosurfactants to treat polluted soils.

## 5. Conclusions

This review has summarised the enhanced hydrocarbon bioremediation studies in cold regions. The oil pollution inevitably raises environmental threats to the ecosystem. Oil spill management via biological treatments can be a promising method to treat polluted soils. Overall, bioremediation approaches are eco-friendly and cost-effective. In situ bioremediation is ideally cost-effective and less invasive to the treatment site, while ex situ bioremediation generates stable and higher removal ability. Among all studies, green technology of phytoremediation is recommended to be an ideal removal treatment due to its various beneficial features, as mentioned previously. This treatment is highly relevant in the Arctic and Alaska, in which different types of plants that can degrade hydrocarbons and grow under low temperatures have been utilised to recover polluted soils. However, phytoremediation is not a one-for-all treatment. Other regions such as Antarctica and Canada that possess limited plant species and minimal infrastructure access are recommended with ex situ bioremediation. Potential molecular and microbial applications were discussed in this review to tackle various limitations. Microbial biosurfactant and immobilisation tools can stimulate the removal efficiency of bioaugmented treatments at low temperatures by enhancing the pollutant bioavailability and maintaining the microbial stability, respectively. The use of recombinant GMMs and engineered enzymes attracts biosafety concerns, while the bioreactor can provide a confined system to remove these foreign agents after treatment. Under cold regions, the bioreactor has been widely applied in the Arctic Circle to remove polluted sewages successfully, while such practice has not been seen in hydrocarbon treatment on soils. The exploration of the bioreactor in treating hydrocarbon-polluted soils at low temperatures can potentially contribute to a more a sustainable treatment for cold regions.

## Figures and Tables

**Figure 1 biology-10-00354-f001:**
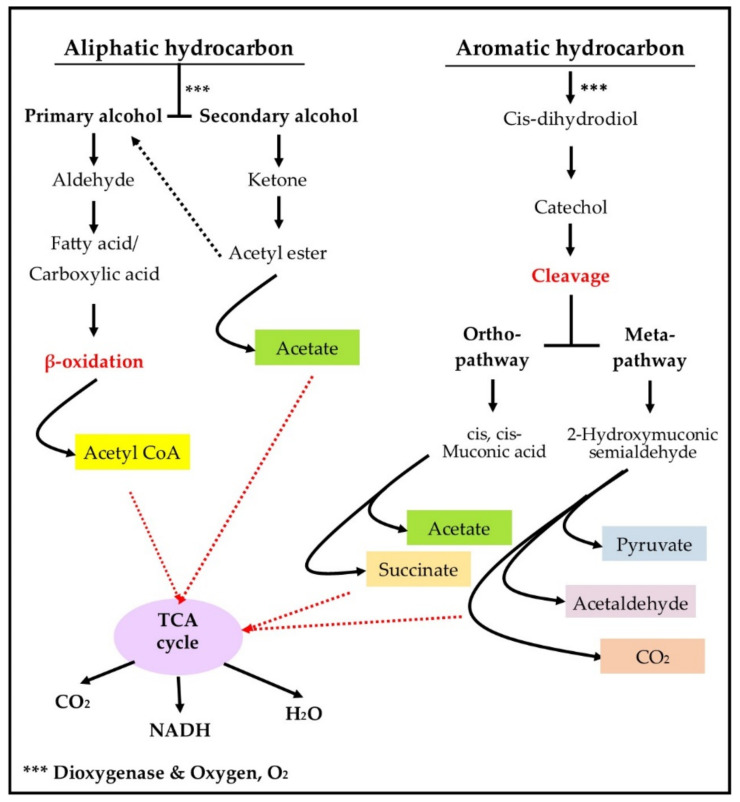
General overview of aerobic biodegradation of hydrocarbons. All end products from aerobic biodegradation will be used in the TCA cycle. Tricarboxylic acid (TCA) cycle is an essential metabolic process that produces important biomass for microbial growth and survival.

**Figure 2 biology-10-00354-f002:**
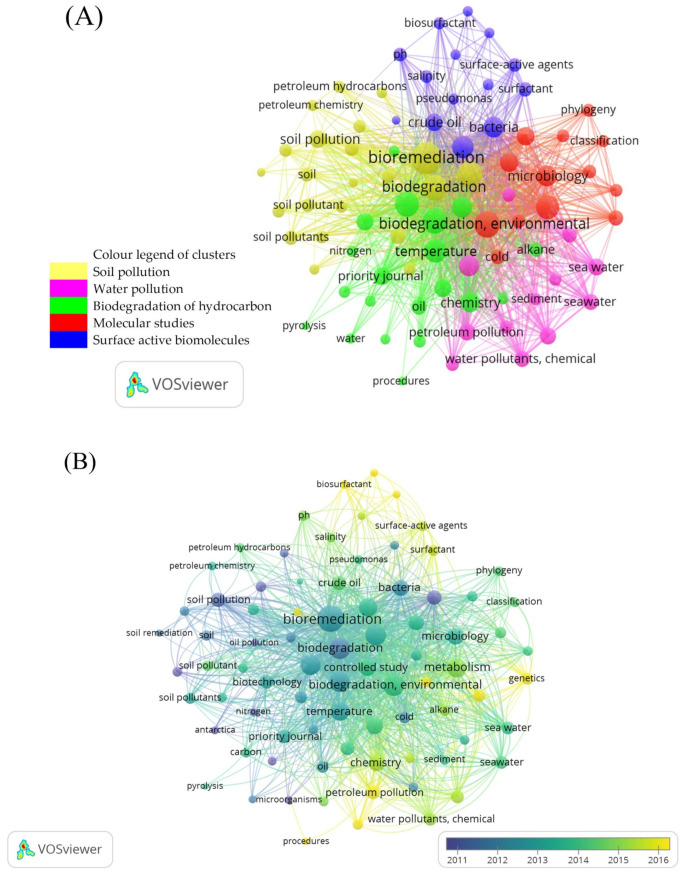
Analysis of keywords. (**A**) Network visualisation of keyword co-occurrence based on total link strength between 90 generated keywords. (**B**) Overlay visual of keyword distribution across average publication year.

**Figure 3 biology-10-00354-f003:**
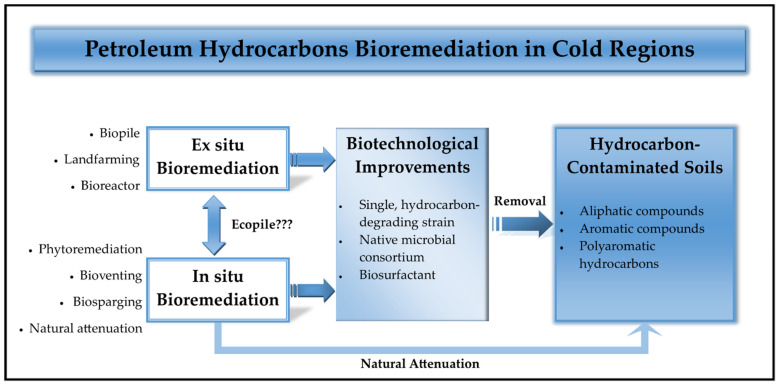
Potential hydrocarbon bioremediation approaches on contaminated soils under cold environments.

**Table 1 biology-10-00354-t001:** Petroleum pollutants generated from oil spillages in cold regions.

Pollutants	Examples	Description	References
Petroleum	Diesel and crude oil	Common oil types found in cold regions that produce toxic effects to the cold ecosystem.	[20,21,22,23,24]
Aliphatic hydrocarbon	n-alkanes (C6–C22)	Major constituents of petroleum oils with linear chain arrangement that can be easily degraded by most of the hydrocarbon-degrading bacteria.	[25,26]
Aromatic hydrocarbon	Benzene, toluene, ethylbenzene, xylene (BTEX) and phenol	Highly volatile, single-ring compounds released by diesel- or petroleum-based products. BTEX compounds are highly unstable and will be readily converted into stable phenolic compounds.	[27,28,29,30]
Polycyclic aromatic hydrocarbon (PAH)	Naphthalene, anthracene, phenanthrene and pyrene	Most recalcitrant pollutants derived from petroleum oils. These compounds contain multiple ring structures that make them highly stable and difficult to be biodegraded.	[31,32]

**Table 2 biology-10-00354-t002:** Potential cold-adapted, hydrocarbon-degrading microorganisms in hydrocarbon bioremediation treatments.

Microorganisms	Origins	Petroleum-Based Contaminants	Removal Efficiency	References
*Arthrobacter* sp. strain AQ5–15 ^A^	King George Island, Antarctica	Phenol	99.4%	[27]
*Rhodococcus* sp. strain AQ5–14 ^A^	King George Island, Antarctica	Phenol	99.1%	[28]
*Pseudomonas* sp. ^A^, *Stenotrophomonas* spp. ^A^ and *Shinella* spp. ^A^	Alpine Binaloud Mountains, Iran	Phenol	99%	[29]
*Sphingomonas koreensis* strain ASU–06 ^A^	Oil-contaminated soil, Egypt	PAHs (Nap, Phe, Ant and pyrene)	98.6%	[31]
*Rhodococcus* sp. strain AQ5–07 ^A^	King George Island, Antarctica	Diesel oil	90.3%	[54,55]
*Pseudoalteromonas* sp. strain P29 ^A^	Arctic marine sediment	Mixed and vacuum crude oil	80–90%	[56]
*Arthrobacter* spp. strains AQ5-05 ^A^ and AQ5-06 ^A^	King George Island, Antarctica	Diesel oil	47.5% (AQ5-06) and 41% (AQ5-05)	[51]
*Dietzia maris* strain NWWC4 ^A^	Subarctic Canada	Arctic diesel	37% ± 6%	[57]
*Ceratobasidum stevensii* strain B6 ^C^ and *Fusarium solani* ^C^	Livingston Island, Antarctica	PAHs (Ant and Phe)	40–89.5%	[32]
*Pseudoalteromonas* spp. ^A^*, Marinobacter* spp. ^A^, *Oleispira* sp. ^A^*, Alcanivorax* sp. ^A^*, Sphingopyxis* sp. ^A^*, Rhodobacter* sp. ^A^ and *Hyphomonas* sp. ^A^	Svalbard, Arctic	Arabian crude oil	17.2–81.9%	[58]
*Rhodococcus* *erythropolis* strain BZ4 ^A^, *Rhodococcus cercidiphyllus* strain BZ22 ^A^, *Arthrobacter sulfureus* strain BZ73 ^A^ and *Pimelobacter simplex* strain BZ91 ^A^	South Tyrol, Italy	Linear, aromatic and polyaromatic hydrocarbons (n-alkanes of C12–C22, phenol, Ant and pyrene)	11–100%	[25]
*Cryptococcus* spp. ^B^, *Candida* spp. ^B^, *Rhodotorula* spp. ^B^, *Mrakia* spp.^B^, *Candida* spp. ^B^, *Cistobasidium* spp. ^B^ and *Pichia* spp. ^B^	King George Island, Antarctica	Linear and aromatic hydrocarbons (Phenol, methanol and n-hexadecane)	13–78%	[26]

^A^ Bacterium. ^B^ Yeast. ^C^ Fungi. Nap: Naphthalene. Ant: Anthracene. Phe: phenanthrene. PAHs: polycyclic aromatic hydrocarbons.

**Table 3 biology-10-00354-t003:** Hydrocarbon bioremediation techniques under cold environments.

Technique	Description	Pros	Cons	References
Phytoremediation	Useful plants are selected and planted at the polluted site	- Highly cost-effective - Green technology - Aesthetic effects - Soil conservation - No supervision needed throughout the treatment	- Slow hydrocarbon attenuation rate - Chance of spreading toxic pollutants into food chain	[50,75,76]
Bioventing/biosparging	Air injection to the soil surface or into deeper soil	- Effective removal on medium molecular weight hydrocarbons - Efficient removal on acclimated soils	- Require optimised and regulated air flow rate - Chance of spreading volatile compounds to air	[53,77,78]
Biopile	Polluted soil is excavated and piled up aboveground with the exploitation of fertilizer, temperature, and irrigation.	- Space-saving - Cost-effective - Efficient for large-scale pollution	- Soil dehydrating - Require continuous electric supply - Soil structure disturbance due to excavation	[15,30,44,49,79]
Landfarming	Contaminated soil is excavated and spread on a treatment bed supplied with tilling system	- Inexpensive - Large treatment capacity - Minimal supervision	- Larger space needed - Time-consuming - Possible leaching of pollutants - Soil structure disturbance due to excavation	[42,45,47,80]
Bioreactor	Polluted soil is excavated into incubation tank supplied with water, oxygen and other requirements.	- Enclosed removal system - Full control on all - bioprocess parameters - Pollutant-specific - Higher removal efficiency - Potential use of genetically modified microorganisms	- High operational cost—Laborious techniques - Different bioreactors for different pollutants - Soil structure disturbance due to excavation	[16,81,82,83,84]

**Table 4 biology-10-00354-t004:** Enhanced in situ bioremediation studies by biostimulation and/or bioaugmentation in cold climate regions.

In Situ Bioremediation	Location	Enhancements	Treatment Period	Removal Efficiency	References
**Phytoremediation**	Subarctic Alaska (PPS)	BST by agricultural fertiliser (20 N: 20 P: 10 K)	Re-examined after 15 years	TPH reduction by 80–95%	[75]
	Sweden (PPS)	BST by 10% *w/w* organic municipal compost	5 months	Removal of 38% (MMW hydrocarbon), 40% (HMW hydrocarbon)	[43]
	Sub-Arctic (DPS)	BST by fertiliser (16.6% N, 4% P and 25.3% K)	330 days	Diesel removal of 97%	[76]
**Bioventing**	New England (PPS)	BST by fertiliser (100 C: 10 N: 1.5 P) + aeration rate at 275 cm^3^/min	12 months	TPH removal of 82.5%	[77]
	Subarctic Macquarie Island (DPS)	BST by N fertiliser (125 mg kg^−1^) + 9 optimised micro-injection (6 mm)	12 months	Removal rate of 1020 mg kg^−1^ per day	[46]

PPS: Petroleum-polluted soil. DPS: Diesel-polluted soil. BST: Biostimulation. C: Carbon. N: Nitrogen. P: Phosphorus. K: Potassium. HMW: High molecular weight. MMW: Medium molecular weight. TPH: Total petroleum hydrocarbon.

**Table 5 biology-10-00354-t005:** Enhanced ex situ bioremediation studies by biostimulation and/or bioaugmentation in cold regions.

Ex Situ Bioremediation	Location	Enhancements	Treatment Period	Removal Efficiency	References
**Biopile**	Antarctica (PPS)	BST by NH_4_NO_3_ and MSP	50 days	Removal of isoprenoid hydrocarbons by 75.8%	[15]
	Antarctica (PPS)	BST by NH_4_NO_3_ and MSP + sunlight (157 h exposure)	2 months	TPH reduction by 75%	[42]
	Canada (PPS)	BST by mature municipal compost and BAT by bacterial consortium	94 days	TPH removal of 74–82%	[36]
	Republic of Ireland (PPS)	BST by fertiliser (25 N: 4 P) and BAT by microbial consortium + phytoremediators	24 months	Below the detectable level with initial TPH concentration of 1613 mg kg^−1^	[92]
**Landfarming**	Sub-Arctic (PPS)	BST by fertiliser (2 MSP: 1 urea) + aggressive tilling	56 days	BTEX and gasoline compounds below the detectable level	[40]
	Italy (PPS)	BST by MPP, MSP, NH_4_Cl and NaCl + periodic tilling and BAT by bacterial consortium	3 months	86% TPH removal	[93]
	Canada (PPS)	BST by fertiliser (100 C: 9 N: 1 P) + 2000 mg kg^−1^ CaCO3 + periodic tilling	2 months	75% TPH removal	[37]
	Canadian Arctic (DPS)	BST by urea and (NH_4_)_2_HPO_4_ + optimised rototilling	3 months	80% TPH removal	[39]
	Korea (PPS)	BST by fertiliser (100 C: 10 N: 1 P) and BAT by oil-degrading microbes	33 days	73.7% TPH removal	[34]

PPS: Petroleum-polluted soil. DPS: Diesel-polluted soil. BST: Biostimulation. BAT: Bioaugmentation. C: Carbon. N: Nitrogen. P: Phosphorus. NH_4_NO_3_: Ammonium nitrate. MSP: Monosodium phosphate. MPP: Monopotassium phosphate. (NH_4_)_2_HPO_4_: Diammonium phosphate. NH_4_CL: Ammonium chloride. NaCl: Sodium chloride. CaCO_3_: Calcium carbonate. TPH: Total petroleum hydrocarbon. BTEX: Benzene, toluene, ethylbenzene and xylenes.

**Table 6 biology-10-00354-t006:** Potential cold-adapted recombinant strain in hydrocarbon bioremediation treatments.

**Targeted Genes**	**Recombinant Strain**	**Temperature**	**Hydrocarbons Nature**	**References**
**Genetically Modified Bacteria**				
*TOL*	*Pseudomonas putida* Q5T	0 °C	Toluene	[109]
*ToMO*	*Pseudoalteromonas haloplanktis* TAC125	15 °C	Derivatives of benzene, phenol, xylene and compounds of toluene, naphthalene	[110]
*DntAaAbAcAd dntB* and *dntD*	*Pseudomonas fluorescens* RE	10 °C	2,4-dinitrotoluene (DNT)	[111]
*BphA* and *bphE*	*Pseudomonas* sp. Cam-10	7 °C	Polychlorinated biphenyls	[108]
**Transgenic Plant**				[112]
*Ntr*	*Nicotiana tabacum*	4 °C	2,4- DNT	

*ToMO:* Toluene-o-xylene monooxygenase. *Dnt:* 2,4-DNT dioxygenase and 4-methyl-5-nitrocathecol monooxygenase. *Bph:* Biphenyl dioxygenase. *Ntr:* Bacterial nitroreductase. *TOL:* A plasmid containing 13 genes including toluate.

## Data Availability

Not applicable.

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
