# Peer review of "Bibliometric Analysis of Hydrocarbon Bioremediation in Cold Regions and a Review on Enhanced Soil Bioremediation"

_biology, 2021, doi:10.3390/biology10050354_

Round 1
Reviewer 1 Report
This is a very well organized and complete work.
Please check for reference citation at the sixth line of paragraph 2.3. Bioremediation Research Trend in Cold Regions.
Author Response
Comment
Please check for reference citation at the sixth line of paragraph. 2.3. Bioremediation Research Trend in Cold Regions.
Answer:
The reference citation has been revised into correct format. Van Eck and Waltman [69]. Line 301

Reviewer 2 Report
Part 2.3 does not provid useful information for research or industry, therefore can be omitted
Referencing to table and figures number should be corrected
Author Response
Comment:
Part 2.3 does not provide useful information for research or industry, therefore can be omitted. Referencing to table and figures number should be corrected
Answer:
Sorry we cannot delete section 2.3 because our review title is “Bibliometric Analysis of Hydrocarbon Bioremediation in Cold Regions and A Review on Enhanced Soil Bioremediation”. Thank you

Reviewer 3 Report
The authors did a comprehensive review on bioremediation of hydrocarbon contaminated soil in cold regions. The information is exhaustive and the manuscript the most important technologies for hydrocarbon remediation. The manuscirpt is worthy to be published after revisions.
Comments
(1) A comparsion of different technologies, showing their advantages and dis-advantages is needed.
(2) Was the author able to do a cost-effective analysis for the technologies?
(3) The authors should provide key chemical reactions leading to the degradation of pollutants by microorganism.
(4) Was the author able to provide the criterion of allowable contents of hydrocarbon in soils?
(5) The author should provide a Table to show the different types of pollutants.
Author Response
Comment 1
A comparsion of different technologies, showing their advantages and dis-advantages is needed.
Answer: The comparison of different techniques have been added. Line 204 Table 3.
Comment 2
Was the author able to do a cost-effective analysis for the technologies?
Answer: Sorry, we are not able to do the cost-effective analysis for the technologies.
Comment 3
The authors should provide key chemical reactions leading to the degradation of pollutants by microorganism.
Answer: The key chemical reaction of hydrocarbon biodegradation is added to enhance the understanding of bacterial biodegradation. Section 2.2. Biodegradation pathway and its metabolic aspects. Line 242 – 294.
Comment 4
Was the author able to provide the criterion of allowable contents of hydrocarbon in soils?
Answer: Sorry, we are not able to provide the criterion of allowable contents of hydrocarbon in soils. It is a good suggestion for the next review paper.
Comment 5
The author should provide a Table to show the different types of pollutants.
Answer: The different types of hydrocarbon pollutants is added. Line 138- 140 (Table1)
